# The Rheology of Polyether Ether Ketone Concentrated Suspensions for Powder Molding and 3D Printing

**DOI:** 10.3390/polym16141973

**Published:** 2024-07-10

**Authors:** Svetlana Yu. Khashirova, Azamat L. Slonov, Azamat A. Zhansitov, Khasan V. Musov, Aslanbek F. Tlupov, Azamat A. Khashirov, Anton V. Mityukov, Alexander Ya. Malkin

**Affiliations:** 1Progressive Materials and Additive Technologies Center, N.M. Bberberov Kabardino-Balkarsky State University, 123, Chernyshevsky Str., 360004 Nalchik, Russia; new_kompozit@mail.ru (S.Y.K.); azamatslonov@yandex.ru (A.L.S.); azamat-z@mail.ru (A.A.Z.); xmusov@gmail.com (K.V.M.); tlupovaslanbek99@gmail.com (A.F.T.); khashaz@yandex.ru (A.A.K.); 2Topchiev Institute of Petrochemical Synthesis, Russian Academy of Sciences, 119919 Moscow, Russia; ant-mityukov@yandex.ru

**Keywords:** PEEK, suspensions, rheology, powder injection molding, 3D printing

## Abstract

The main goal of the work was to use rheological methods for assessing the properties of a composition based on polyether ether ketone (PEEK) to determine the concentration limits of the polymer in the composition and select the optimal content of this composition for powder molding. The rheological properties of highly filled suspensions based on PEEK and paraffin, as well as in paraffin–polyethylene mixtures at various component ratios, were studied. These materials are designed for powder injection molding and 3D printing. Suspensions with a PEEK powder content above 50% are not capable of flow and, with increasing pressure, slide along the surface of the channel. For compositions with a higher content (60 and 70 vol.%) PEEK, independence of the storage modulus from frequency is observed, which is typical for solids and confirms the assignment of such suspensions to elastic–plastic media. The introduction of high-density polyethylene into the composition helps improve the technological properties of suspensions, expanding the range of fluidity, although it leads to an increase in viscosity. In suspensions with a mixed composition of the liquid phase, with increasing temperature, a decrease in the storage modulus is observed at 120 °C and, on the contrary, an increase at 180 °C. The latter may be a consequence of the evaporation of paraffin and the softening of PEEK due to the approach to the glass transition temperature of the polymer. Suspensions with 40% PEEK content have an optimal set of rheological properties for powder injection molding. A 3D printing filament was also obtained from a composition with 40% PEEK, which had good technological properties for FDM 3D printing. Products of satisfactory quality from suspensions with 50% PEEK can be produced by powder injection molding, but not by 3D printing. The selected compositions were used to obtain real PEEK products for practical applications.

## 1. Introduction

Polyether ether ketone (PEEK) is one of the most promising modern engineering plastics with high strength and performance over a wide temperature range. A large number of studies have been devoted to it, including work on its synthesis [1,2] and the development of composite materials based on it [3,4,5]. PEEK, having a complex of high thermal and mechanical properties, as well as resistance to aggressive environments and radiation, is used in various fields, including such areas as the aerospace and biomedical industries [6,7,8].

PEEK, as a thermoplastic polymer, can be (and is actually) processed using traditional extrusion and injection molding methods [5]. Also, PEEK is in demand in various additive technologies, including selective laser sintering (SLS) and the filament deposition method (FDM) [9,10,11,12,13,14,15]. The use of PEEK in SLS technology is accompanied by the formation of a large amount of waste, since the powder used serves as a support for constructing the model. Keeping the powder in a high-temperature chamber can lead to a decrease in properties, but it is possible to reuse the powder, including for highly filled suspensions [16,17].

However, the high melting point of PEEK (343 °C) requires the use of special equipment, which is not always available and, in general, significantly increases technological costs, especially when it comes to small portions of specific products with complex configurations. Therefore, it is important to use other technological solutions, which include powder injection molding and 3D printing of highly concentrated suspensions, which can dramatically reduce the temperature of processing and reduce the cost of manufacturing the mold (for casting).

The use of the methods of additive technologies requires the creation of suspensions based on low-viscosity liquids (matrices), which are the primary working material when molding products. Accordingly, the rheological properties of these suspensions determine the features and boundaries of the technological process and therefore, the measurement of these properties is the initial step in optimizing the composition of the suspensions for the implementation of the powder injection molding process. Technologically, it is advisable to use suspensions with the maximum possible content of solid particles (often called feedstock, as is customary in powder metallurgy, which is the ancestor of the powder casting method).

General concepts about the rheological properties of highly filled suspensions have been formulated in publications [18,19]. As it has been shown, the factor that determines the limiting degree of filling, φ* is the size of particles in the dispersed phase [19], on which the characteristic concentration boundaries of the areas of rheological state of suspensions also depend on fluidity and elastic-plasticity, in which the powder casting of suspensions is possible [20]. This applies not only to traditional materials such as metal and ceramic powders, but also to high-temperature engineering thermoplastics, which are the subjects of consideration in this work.

The involvement of structural heat-resistant polymers in the technological process of powder casting and 3D printing has great prospects for the further development of powder technology. In this regard, the present work was undertaken. Its goal is to select optimal binder compositions based on the results of rheological studies, suitable for both powder casting and 3D printing, which will make it possible to obtain highly filled PEEK compositions in relation to the implementation of the process of powder injection molding, the extrusion of filaments for 3D printing using the FDM method, and ultimately the production of specific products. Thus, the goal of the work was to use rheological methods to create a composition based on PEEK for powder molding and printing, which would allow us to produce real technical products for real technological applications and illustrate the achievement of this goal by obtaining such products.

## 2. Materials and Methods

The synthesis of PEEK was carried out by high-temperature polycondensation using the nucleophilic substitution mechanism by reacting hydroquinone and 4,4′-difluorobenzophenone in the presence of potassium carbonate and sodium carbonate in diphenyl sulfone in argon. The reaction mixture was slowly heated to 320 °C and kept at this temperature for 5 h, after which it was unloaded onto a metal tray. The choice of this temperature was determined by achieving the PEEK glass transition region (see thermogram below), which resulted in increased molecular mobility of macromolecules, which ensured the possibility of powder compaction.

The cooled polymer layer was crushed and washed with hot acetone and hot distilled water. Then, the polymer was dried at a temperature of 120 °C in a vacuum for 12 h.

The IR spectra of the obtained sample, used in further studies, for two frequency ranges are presented in Figure 1.

IR spectra were recorded on a Fourier spectrometer (Spectrum Two; PerkinElmer, Inc., Waltham, MA, USA) in the range of 4000–450 cm^−1^ with a spectral resolution of 0.4 cm^−1^.

The resulting spectra confirm the structure of the PEEK sample used. All absorption bands characteristic of PEEK are observed in the spectra. A sharp peak in the region of 1646 cm^−1^ is associated with vibrations of the C=O group, peaks in the region of 1300–1100 cm^−1^ are associated with vibrations of C-O-C bonds. Small bands with peaks in the region of 3040–3062 cm^−1^ correspond to vibrations of C-H bonds. Intense peaks at 1592 and 1489 cm^−1^ are attributed to the C=C vibrations of the aromatic ring.

The thermal characteristics of the sample are as follows: The melting point during the initial test is 346 °C, and 341 °C at the second test.

The results of thermogravimetric analysis are shown in Figure 2.

According to these data, the following indicators characterize the heat resistance of PEEK: the loss temperatures of 2%, 5% and 10% of the mass are 547, 558 and 567 °C, respectively.

The phase transition temperature of the synthesized polymers was determined by differential scanning calorimetry (DSC) on a DSC 4000 instrument (PerkinElmer, Inc., Waltham, MA, USA). The quantity of 5–10 mg of the sample was sealed in an aluminum hermetic pan and heated to 250 °C at 10 °C/min under a flow of nitrogen (20 mL/min).

The results of DSC analysis are shown in Figure 3, where endo- and exothermic temperatures corresponding to the phase transition of the PEEK crystals are clearly visible.

The viscosity properties of PEEK in the temperature range at which PEEK can be processed using traditional methods are shown in Figure 4. The experiments were carried out on a two-capillary rheometer, which made it possible to exclude entrance correction by comparing the results obtained on capillaries of different lengths. When processing the initial measurement results, we also introduced a correction for the non-Newtonian nature of flow by using the Rabinowitsch–Weissenberg method.

The binders used were food-grade highly purified paraffin grade P-2 (designated below as P-2). Its standard melting point is 52–56 °C, and its main substance content is no less than 99%. Also, high-density polyethylene (LDPE), grade 15813-020, produced by SIBUR Co(Moscow, Russia) (designated below as PE) was used. The melt index of this PE is 2.0 g/10 min at 190 °C and at a load of 21.2 N. The studies of rheological properties were carried out on a capillary rheometer LCR-7001 from Dynisco (Franklin, MA, USA) and a rotational rheometer RS-600 (ThermoHaake, Waltham, MA, USA) at different given shear rates.

The particle size distribution of PEEK was determined using an Analysette 22 laser diffractometer from Fritsch (Idar-Oberstein, Germany) and a Vega 3 scanning electron microscope (SEM) from Tescan (Brno, Czech Republic).

Injection molding of the samples was performed on an SZS-20 injection molding machine (Haitai Machinery, Zhejiang, China). Compositions with P2-PE mixtures were obtained using a desktop twin-screw micro extruder PJSZ (Haitai Machinery, Zhejiang, China) with L/D = 30 at temperatures by zone: T1 = 105 °C, T2 = 125 °C, T3 = 140 °C, T4 = 150 °C, T5 = 140 °C.

## 3. Results and Discussion

The particle size of the suspension is an important factor in determining the optimal concentration range used in powder casting. A study of the granulometric composition of the synthesized PEEK showed that the powder has a narrow particle size distribution with an average size of about 80 μm (Figure 5). In this case, the particles are characterized by a shape close to spherical. It should be noted that large particles may be agglomerates of smaller particles.

According to [19], such powders belong to the group of macroparticles and in the absence of any specific factors, the maximum degree of volume-filling can reach about 0.5 (50 vol.%).

Suspensions were obtained based on P2 and PEEK, which was introduced in amounts of 20, 30, 40, 50 and 60 vol.%. To accomplish this, P2 was heated to 100 °C and then mixed with the required amount of polymer powder.

Figure 6 shows that when trying to introduce 60% PEEK into paraffin, the amount of paraffin, which should form a continuous matrix, is not sufficient to wet the PEEK particles, making it impossible to prepare a suspension. In this regard, the maximum concentration of powder in the studied suspensions was approximately 50%. This is consistent with the maximum filling degree mentioned above.

Figure 7a shows the pressure dependencies on the specified volume flow rate measured on a capillary viscometer, recalculated using standard Poiseuille flow formulas into the dependence of the apparent viscosity on the apparent shear rate (without taking into account the correction for the non-Newtonian behavior of suspensions). The experiment was carried out at 50 °C, i.e., slightly lower than even the melting point of P2. This means that, actually, plastic deformation occurred, rather than viscous flow.

At first glance, according to Figure 7a, it seems that we are dealing with ordinary non-Newtonian power-law-type fluids. However, in reality, this is not the case. Figure 7b shows that, in reality, only 20% and 30% of the suspensions behave like liquids (i.e., can flow), while suspensions with higher concentrations of PEEK are characterized by a constant stress, at which they are forced through the viscometer channel regardless of speed. This reflects the mechanism of sliding of a highly filled suspension. Therefore, the values η along the ordinate do not in any way represent “viscosity”. It is significant that the transition from 40% to 50% suspension leads to a significant increase in stress and in pressure, which is necessary to fill the mold.

The flow curves of PEEK dispersions in the region of stable flow are described by a standard power-law-type equation with an index of the dependence of the apparent viscosity on the shear rate of the order of −0.73. The temperature dependence of the viscosity (activation energy of viscous flow) depends on the choice of the shear rate at which the comparison is made. In the region of lower rates, it is close to 100 kJ/mol, but significantly decreases with increasing shear rate.

From the prepared suspensions of PEEK in a P2 matrix, samples of bars were obtained by injection molding (Figure 8). A comparison of the quality of the resulting products showed that the most suitable composition was a suspension containing 40 vol.% PEEK. With a lower PEEK content, the resulting samples are characterized by increased fragility, which makes it impossible to remove them from the mold. With an increased PEEK content (up to 50 vol.%, which corresponds to the limit of filling) the viscosity is too high, which makes it impossible to obtain intact samples.

To give the samples additional strength and ensure their integrity during the sintering process, the composition of the binder was changed, and matrixes were obtained with the addition of PE to P2. The rheological properties of these compositions were studied on a rotational rheometer at low shear rates (Figure 9) and on a capillary viscometer at a wide range of high shear rates (Figure 10).

The data presented in Figure 9a show some increase in viscosity due to the introduction of PE, although the effect is not very significant. However, the most interesting and unexpected observation is presented in Figure 9b—the dilatant increases in viscosity with increasing stress, which takes place in the region of low shear rates. This “strange” result means that during the measurements, these samples did not demonstrate a real bulk flow, but more likely, with an increase in the specified shear rate, flow stratification (shear banding) took place and the obtained values of apparent viscosity did not correspond to the value of the bulk viscosity of the suspension. In the region of shear rates above 1 s^−1^, the flow became unstable, and the apparent viscosity was the viscosity of the P2 layer.

Experimental data presented in Figure 10 reflect the results of measurements performed in the region of higher shear rates on a capillary viscometer.

A comparison of the data presented in Figure 7b and Figure 10b shows that the increase in viscosity with increasing PE content in the binder is more pronounced here than at low shear rates in suspensions in pure P2. In addition, it is obvious that the apparent viscosity values in Figure 9 and Figure 10 do not fit together. This is due to the different nature of the flow, since at high shear rates flow does occur, although with a strong scatter of experimental points due to the mixing of components at high stresses. However, the introduction of PE into the matrix allows the suspension to flow, which is beneficial for improving the technological properties of the suspension.

Also (and this is quite natural) an increase in the content of PEEK powder leads to a significant increase in viscosity, despite adjusting the matrix composition (Figure 11). This figure presents data for different matrix compositions and PEEK contents. Figure 11a shows the direct results of measurements on a capillary viscometer, which look like ordinary flow curves. However, a calculation of the dependence of the apparent viscosity on shear stress shows that suspensions with 60% and 70% PEEK do not flow, but slide along the channel wall at a constant stress regardless of speed—the so-called spurt occurs. Only the 50% suspension can flow. This result is consistent with the above estimate of the filling limit for suspensions with a selected filler particle size. However, it is interesting to note that the use of PE in a mixture with P2 makes it possible to extrude a composition even with 70% PEEK and obtain filaments for 3D printing.

Additional information about the physical state of suspensions of various compositions, explaining the behavior of these materials under shear deformation (flow or sliding), was obtained from studies of their viscoelastic properties.

A study of viscoelastic properties of suspensions containing 50% PEEK with different ratios of P2 and LDPE in the matrix showed that at 120 °C, an increase in the P2 content leads to an increase in the storage modulus until solid-state behavior is achieved (Figure 12). With a P2:PE ratio of 60:40, the storage modulus is practically independent of frequency. This effect is due to the phase state of the dispersion medium at this temperature. P2, with a melting point of about 55 °C at the test temperature, is a liquid and can dissolve (form a thermodynamically compatible mixture) in PE [21], which is also practically in a melted state (slightly above the melting point). However, with an increase in the proportion of PE in the matrix, the storage modulus of the composition decreases, since the storage modulus of PE is lower than PEEK and its contribution to the elasticity of the mixture reduces the elasticity of the composition.

However, at 180 °C, when PEEK softens as a result of exceeding its glass transition temperature, the elasticity decreases for compositions containing more P2.

Viscoelastic properties depend on the concentration of the solid phase (PEEK). This is clearly visible in Figure 13, and the most interesting effect is a sharp jump in the elastic modulus when the critical filling threshold is exceeded (about 50%). And, if at lower concentrations suspensions behave like typical viscoelastic liquids, then at higher concentrations (in our case 60% and 70%), the storage modulus does not depend on frequency. The latter is direct evidence that at these concentrations, highly filled suspensions are solid-like media.

Simultaneously with the storage modulus (Figure 12 and Figure 13), the frequency dependence of the loss modulus was also measured. Table 1 summarizes the measured viscoelastic properties of the four suspensions studied.

The results of the measurements of the viscoelastic properties of the studied compositions, in addition to the data on the viscometric properties of the suspensions discussed above, allow us to conclude that concentrations of 40 and 50% are the most acceptable for obtaining products. However, powder molding technology requires a higher content of the target component. Therefore, to control the conclusion made about the most acceptable range of concentrations, compositions with 60 and 70 vol.% PEEK were used for injection molding of standard samples. Casting was carried out at a temperature of the material cylinder of the injection machine equal to 210 °C and a mold temperature of 90 °C. This temperature lies above the melting point of the binder, so the choice of this temperature ensures a stable extrusion process.

From a composition containing 70 vol.% PEEK, it was not possible to mold the product at all due to the solid-like state of the suspension. In the case of a suspension with 60 vol.%, samples of satisfactory quality were obtained (Figure 14). However, as experiments have shown, this composition is not entirely suitable for producing products with more complex geometries, compared to a suspension of the optimal composition with 50% PEEK.

Meanwhile, the use of the developed compositions for additive technology, in particular for 3D printing using the FDM method, imposes additional restrictions on the composition of suspensions. Thus, from a composition with 50% PEEK content it is quite possible to obtain filaments with a diameter of 1.75 mm intended for 3D printing. However, it turned out that all compositions with 50% PEEK and different ratios of components in the binder had too high of a fragility. This makes it impossible to use them in the FDM method. A filament was successfully obtained from a suspension containing 40% PEEK (Figure 15a). Testing of this filament in 3D printing showed the possibility of producing a product from it at a temperature of 240 °C (Figure 15b).

## 4. Conclusions

A study of the rheological properties of highly filled suspensions based on PEEK with a paraffin matrix showed that an increase in the volume content of PEEK powder in the suspension leads to a natural increase in viscosity. When the concentration reaches approx. 50% or higher, the flow of the suspension becomes impossible and, with increasing pressure, wall-sliding occurs. The concentration range of about 50% corresponds to the maximum degree of filling for particles of the used size. At PEEK concentrations of 60% and 70%, the storage modulus does not depend on frequency. This proves that such suspensions are solid-like media, do not have fluidity, and cannot be used for powder molding. The addition of PE to obtain highly filled suspensions with low fragility showed that, with an increase in its content, the viscosity increases. At the same time, the addition of PE leads to a decrease in the storage modulus at 120 °C and an increase at 180 °C. The latter may be due to the evaporation of paraffin and softening of PEEK due to the transition of the amorphous phase through the glass transition temperature. Thus, the optimal complex of rheological and mechanical properties is achieved when using a suspension with 40% PEEK, which has sufficient fluidity for casting and strength for removing the product from the mold. With this PEEK content, a filament for 3D printing was also obtained, which turned out to be suitable for producing a product using the FDM method. Compositions with 50% PEEK could also be used for powder molding, but they are too brittle for 3D printing.

## Figures and Tables

**Figure 1 polymers-16-01973-f001:**
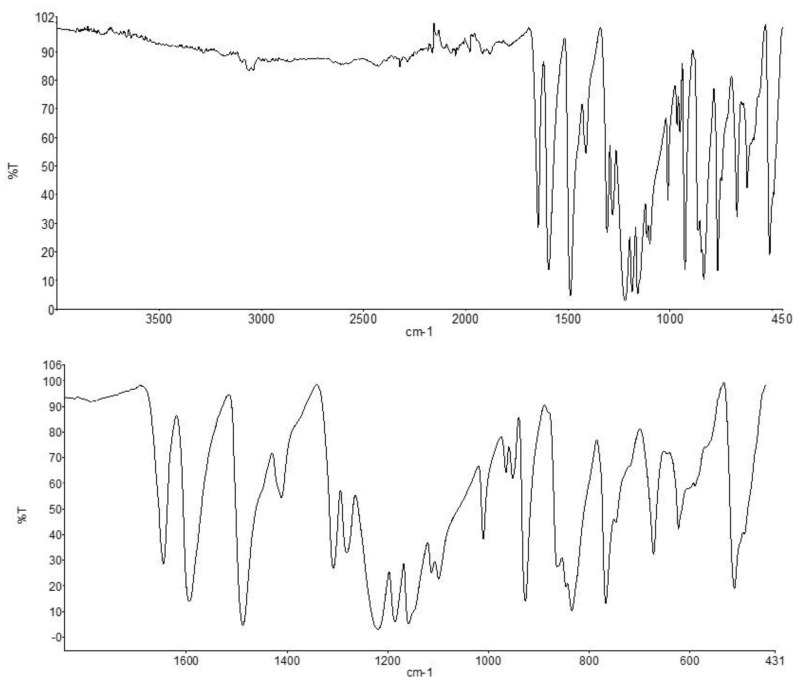
IR spectra of the PEEK sample used in this study.

**Figure 2 polymers-16-01973-f002:**
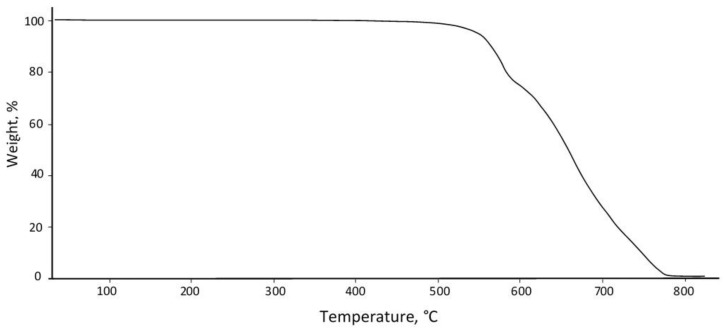
Results of thermogravimetric analysis of PEEK.

**Figure 3 polymers-16-01973-f003:**
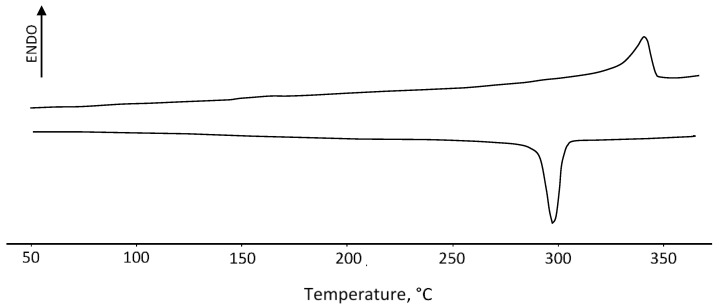
DSC diagram of the PEEK sample used.

**Figure 4 polymers-16-01973-f004:**
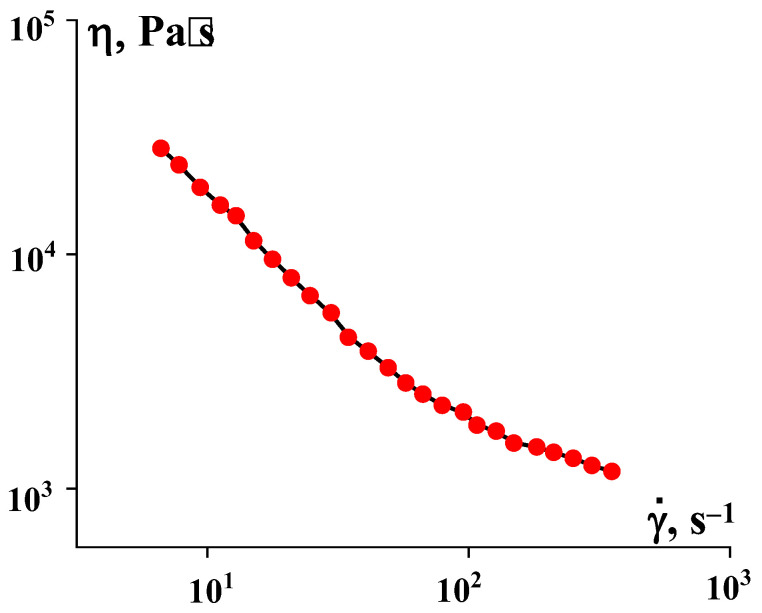
PEEK flow curve at 380 °C (capillary viscometry data).

**Figure 5 polymers-16-01973-f005:**
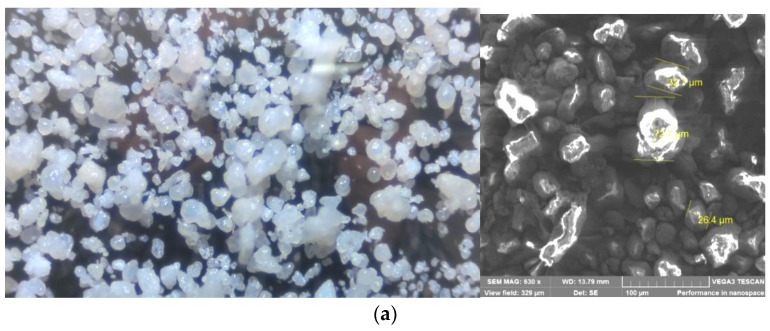
Microphotographs (**a**) and particle size distribution curve (**b**) of PEEK powder.

**Figure 6 polymers-16-01973-f006:**
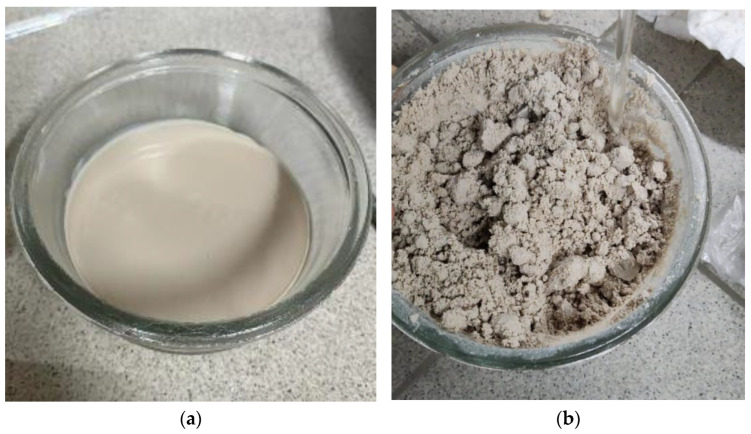
PEEK suspensions in paraffin: (**a**) 20 vol.%; (**b**) 60 vol.%.

**Figure 7 polymers-16-01973-f007:**
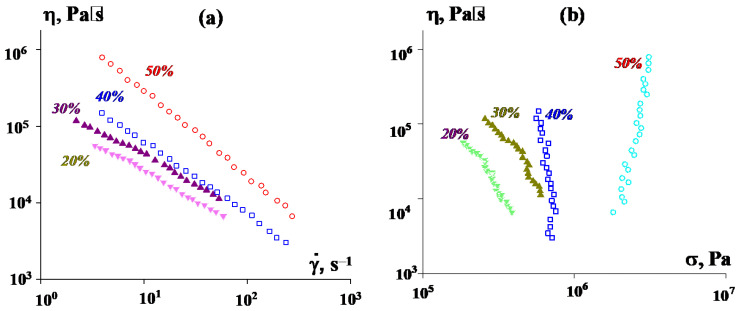
Apparent viscosity of PEEK suspensions of various concentrations (indicated in the curves) depending on the apparent hear rate (**a**) and shear stress (**b**).

**Figure 8 polymers-16-01973-f008:**
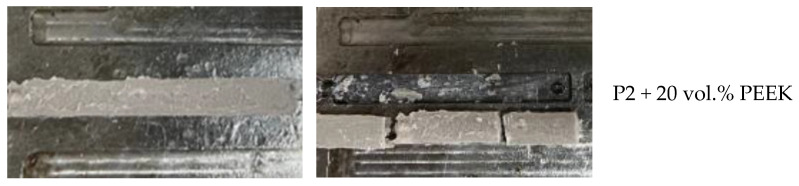
Molded samples from paraffin-based suspensions before (**a**) and after (**b**) removal from the mold.

**Figure 9 polymers-16-01973-f009:**
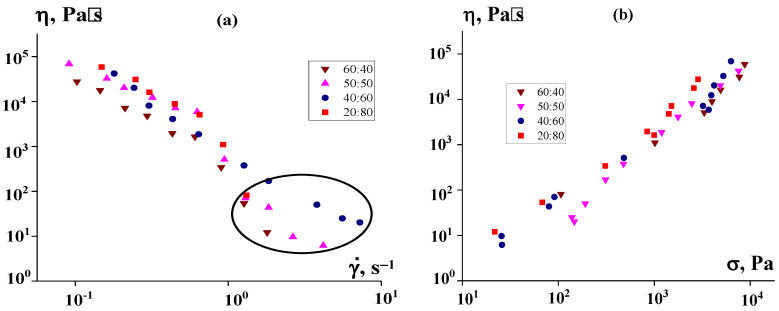
Flow curves of PEEK suspensions in a P2:PE mixture at various component ratios (indicated on the curves) obtained on a rotational rheometer. PEEK concentration was 50%. The results are presented as the dependence of the apparent viscosity on the shear rate specified in the experiment (**a**) and on the recalculated shear stress (**b**). The oval shows the area where instability occurs. The temperature was 120 °C.

**Figure 10 polymers-16-01973-f010:**
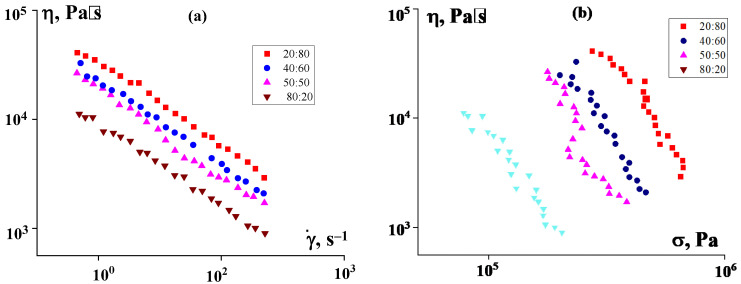
Flow curves of PEEK suspensions in P2-LDPE mixtures at various component ratios (indicated on the curves), obtained on a capillary rheometer in the region of high shear rates. PEEK concentration was 50%. The results are presented as the dependence of the apparent viscosity on the shear rate specified in the experiment (**a**) and on calculated shear stress (**b**). The temperature was 120 °C.

**Figure 11 polymers-16-01973-f011:**
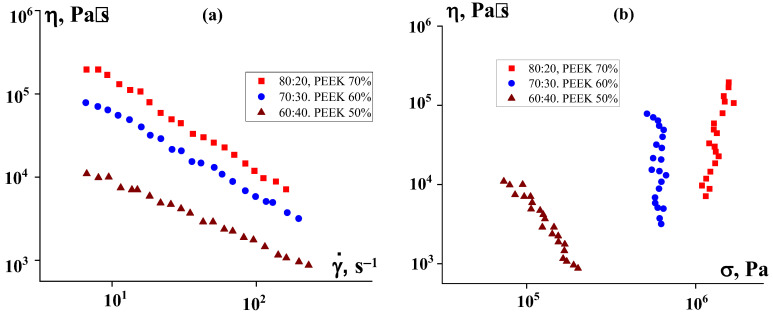
Dependences of apparent viscosity on shear rate (**a**) and shear stress (**b**) for suspensions with different PEEK contents and different ratios of P2 and PE at 120 °C.

**Figure 12 polymers-16-01973-f012:**
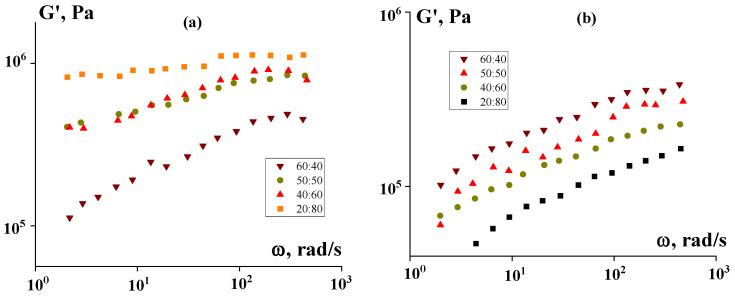
Dependence of the storage modulus on frequency for suspensions containing 50% PEEK and different ratios of P2: PE (indicated in the figure) at 120 °C (**a**) and 180 °C (**b**).

**Figure 13 polymers-16-01973-f013:**
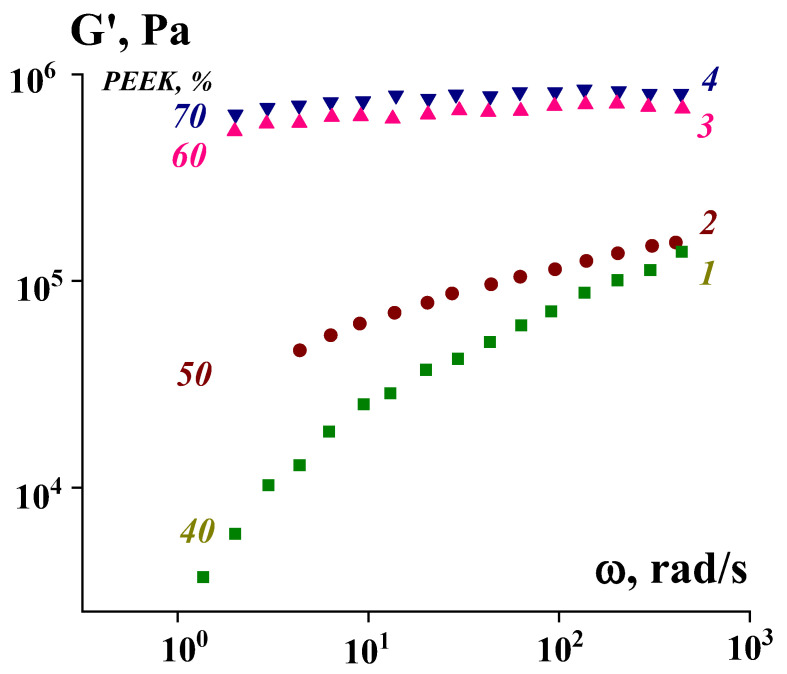
Frequency dependencies of the storage modulus of compositions with different PEEK content and different P2:PE ratios—30:70 (1), 60:40 (2), 70:30 (3), and 80:20 (4) at 180 °C.

**Figure 14 polymers-16-01973-f014:**
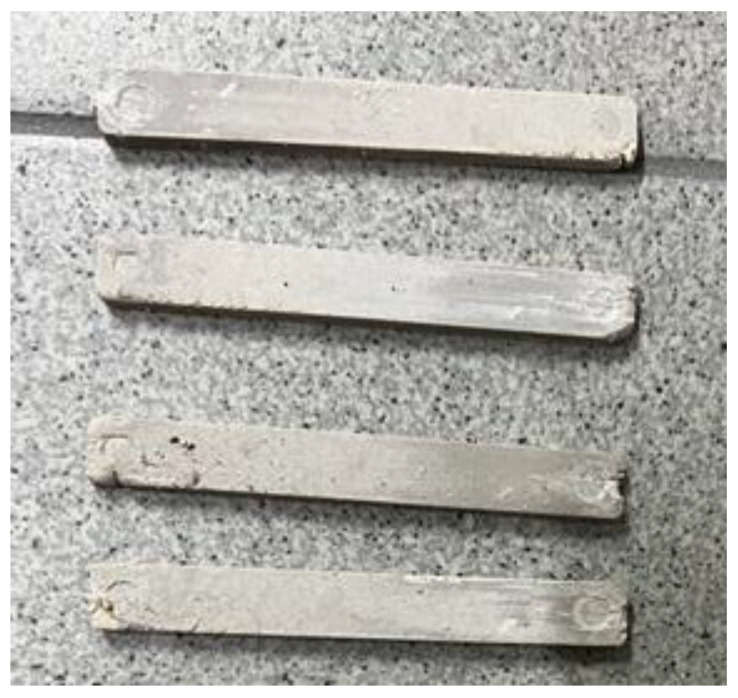
Samples obtained from a suspension of the composition at 60% PEEK and the ratio of binder components P2:PE = 70:30.

**Figure 15 polymers-16-01973-f015:**
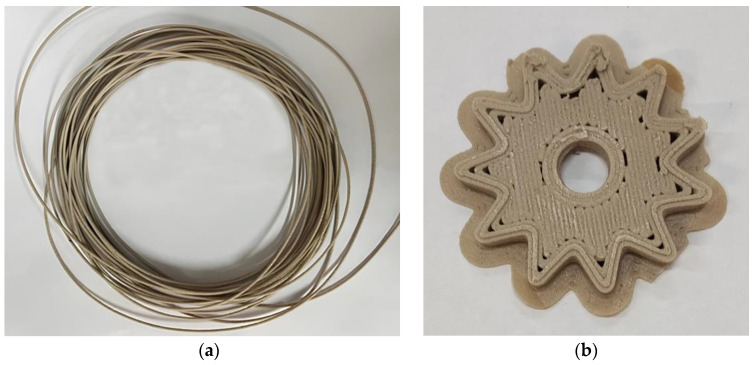
Filament containing 40 vol.% PEEK (**a**) and a product printed from it (**b**).

**Table 1 polymers-16-01973-t001:** Relationships between the storage and loss components of the elastic modulus for different compositions.

Composition	Relationship between G′ and G″	
	120 °C	180 °C
40% PEEK P2:PE = 30:70	G′ = G″	G″ > G′
50% PEEK P2:PE = 20:80	G′ > G″	
50% PEEK P2:PE = 40:60		
50% PEEK P2:PE = 50:50		
50% PEEK P2:PE = 60:40		G′ = G″
60% PEEK P2:PE = 70:30	G′ > G″; G′(ω) = const	
70% PEEK P2:PE = 80:20		

## Data Availability

Data are contained within the article.

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
