# Peer review of "The Rheology of Polyether Ether Ketone Concentrated Suspensions for Powder Molding and 3D Printing"

_polymers, 2024, doi:10.3390/polym16141973_

Round 1
Reviewer 1 Report
Comments and Suggestions for Authors
(1) Abstract is missing the problem statement, why authors have carried out this study, what was lacking in the previous literature which authors have tried to cover here?
(2) In the end of the abstract, there should be a sentence describing the practical utilization of the study in some field.
(3) The problem statement in the introduction section is not clear. Please emphasize the problem statement in the introduction.
(4) Do you have any possible explanation for the statement “The reaction mixture was slowly heated to 320 °C …” in the materials and methods section?
(5) Two corrections are commonly applied for the capillary die measurement in order to obtain the correct viscosity. They are the Bagley correction and the Weissenberg-Rabinowitsch correction. Specific information on such corrections related to your study should be provided.
(6) What is the recommended shear rate and viscosity range of this study based on previous literature?
(7) A discussion on flow behaviour index and flow activation energy is missing. Specific information on these should be provided.
(8) To further improve the rheological properties, the authors may carry out experiments with alternate binder materials. An investigation into alternative binder materials may result in enhanced final product characteristics.
(9) In line 296, the chosen mould temperature was 90 °C. What is the explanation behind selecting such a temperature?
(10) There are some typos scattered throughout the paper, authors are suggested to revise the paper carefully and correct them.
Author Response
Dear Reviewer,
I am extremely grateful to you for your careful consideration of the manuscript
and the comments made. Correcting the shortcomings you noted would undoubtedly
improve the quality of the article.
Our corrections to the text, made in accordance with your comments, are marked
below and in the text with a yellow background.
(1) Abstract is missing the problem statement, why authors have carried out this study, what
was lacking in the previous literature which authors have tried to cover here?
Thank you for the comment. The following insert was added to Abstract
The main goal of the work was to use rheological methods for assessing the properties of
a composition based on PEET to determine the concentration limits of the polymer in the
composition and select the optimal content of this composition for powder molding.
(2) In the end of the abstract, there should be a sentence describing the practical utilization
of the study in some field.
Thank you for the comment. The following insert was added to Abstract
The selected compositions were used to obtain real PEEK products for practical
applications.
(3) The problem statement in the introduction section is not clear. Please emphasize the
problem statement in the introduction.
Thank you for the comment. Thai had been made and the following statement hs been
added
Thus, the goal of the work was to use rheological methods to create a composition based
on PEEK for powder casting and printing, which allowed us to produce real technical
products for real technological applications and illustrate the achievement of this goal by
molding such products
(4) Do you have any possible explanation for the statement “The reaction mixture was slowly
heated to 320 °C …” in the materials and methods section?
Thank you for this comment. The following explanation was added
The choice of this temperature was determined by achieving the PEEK glass transition
region (see thermogram below), which resulted in increased molecular mobility of
macromolpeculbs, which ensured the possibility of powder compaction
(5) Two corrections are commonly applied for the capillary die measurement in order to
obtain the correct viscosity. They are the Bagley correction and the WeissenbergRabinowitsch correction. Specific information on such corrections related to your study
should be provided.
Yes, both corrections were applied. The following text explains thist.
The experiments were carried out on a two-capillary rheolmeter, which made it possible
to exclude entrance correction by comparing the results obtained on capillaries of different
lengths. When processing the initial measurement results, we also introduced a correction
for the non-Newtonian nature of floe by using the Rabinowitsch-Weissenberg method.
(6) What is the recommended shear rate and viscosity range of this study based on previous
literature?
We are not aware of any literature data that would give recommendations on the level
of viscosity and recommended shear rates for powder molding of PEET-based
composites. Therefore, such data were established in this work, and just this was one of
its tasks.
(7) A discussion on flow behaviour index and flow activation energy is missing. Specific
information on these should be provided.
Answering this comments, we added the following addition to the tex
The flow curves of PEEK dispersions in the region of stable flow are described by a
standard power-law type equation with an index of the dependence of the apparent
viscosity on the shear rate of the order of -0.73. The temperature dependence of the
(activation enegrgy of viscous flow) viscosity depends on the choice of shear rate. at
which comparison is made. In the region of low rates it is close to 100 kJ/mol, but
significantly decreases with increasing shear rate
(8) To further improve the rheological properties, the authors may carry out experiments with
alternate binder materials. An investigation into alternative binder materials may result
in enhanced final product characteristics.
Thank you. It's a good suggestion. In our further research, we plan to vary the composition
of the binders, but this is a separate large work that now goes beyond the scope of this
study.
(9) In line 296, the chosen mould temperature was 90 °C. What is the explanation behind
selecting such a temperature?
Thank you for the comment. The explanation is as follows.
This temperature lies above the melting point of the binder, so the choice of this
temperature ensures a stable extrusion process
(10) There are some typos scattered throughout the paper, authors are suggested to
revise the paper carefully and correct them.
Thanks to the reviewer for carefully reading the manuscript of our article. We read the text
again and did our best. to get rid of errors and typos

Reviewer 2 Report
Comments and Suggestions for Authors
This work investigates the rheology of mixtures containing poly (ether ether ketone) (PEEK) and paraffin, as well as paraffin-polyethylene blends, at different ratios for powder injection molding and 3D printing. It uniquely explores the optimal PEEK loading percentage for successful 3D printing by incorporating flow modifiers and binders such as polyethylene (PE) and paraffin. The authors compared the viscosity and rheological performance of different blends and drew meaningful conclusions. Mixtures with 40% PEEK have the best flow properties for powder injection molding. A filament with 40% PEEK showed good performance for FDM 3D printing. Products with 50% PEEK can be successfully made using powder injection molding but not with 3D printing.
The manuscript requires significant language revisions; in several places, the English used by the author is not purely scientific and appears more casual. The reviewer encourages the authors to work on the language, especially in the results and discussion section, to make it more scientific and impactful. Here are the comments:
1. In the manuscript title, the spelling of "PEEK" or "polyether ether ketone" is incorrect.
2. The introduction needs to address the challenges associated with PEEK materials, including how selective laser sintering (SLS) and 3D printing of PEEK generate waste, and the effects of thermal history and rheological properties during the 3D printing process on the processability of PEEK. Additionally, explores the potential of these suspension blends to improve the recycling of PEEK powder generated from SLS/3D printing.
3. Line 59: "As has been shown" lacks a subject, suggesting that "it" is missing.
4. Paragraph 4, line 65, does not end with a full stop.
5. Paragraph 5 needs to be improved. It contains a run-on sentence, and the study's objective is not clearly defined for the readers.
6. In the materials/methods section, the authors have not mentioned the analysis methods and conditions for FTIR, TGA, and DSC, which would aid reader understanding.
7. Line 123: There seems to be extra spacing after the full stop.
8. Line 160: The abbreviation "app." for approximately is not defined earlier. All abbreviations should be defined upon their first appearance.
9. Lines 175-176: The sentence could be rewritten, as starting a formal sentence with "But" in scientific articles is incorrect. Try using words like "however" or "nevertheless."
10. Lines 180-181 need to be rewritten to sound more scientific.
11. Figure 8b must be reinserted as it is an incomplete image.
12. Lines 195-199 are right-aligned; the text should be adjusted for improved distribution.
13. In Figure 10(b), the legend color (light blue) does not match the graph legend notation (brown) for the 80:20 blend.
14. Lines 263-265 need language improvement. "The picture reverses" does not fit in the scientific explanation.
Comments on the Quality of English LanguageRequires substantial language adjustments and thorough proofreading to enhance its scientific impact for readers.
Author Response
Dear Reviewer,
I am extremely grateful to you for your careful consideration of the manuscript and the comments made. Correcting the shortcomings you noted would undoubtedly improve the quality of the article.
Our corrections to the text, made in accordance with your comments, are marked in the text with a gray background.
1. In the manuscript title, the spelling of "PEEK" or "polyether ether ketone" is incorrect.
Thank you.
Corrected.
2. The introduction needs to address the challenges associated with PEEK materials, including how selective laser sintering (SLS) and 3D printing of PEEK generate waste, and the effects of thermal history and rheological properties during the 3D printing process on the processability of PEEK. Additionally, explores the potential of these suspension blends to improve the recycling of PEEK powder generated from SLS/3D printing.
Thank you for this suggestions. We’ve added some text and additional references.
3. Line 59: "As has been shown" lacks a subject, suggesting that "it" is missing.
Thank you.
Corrected.
4. Paragraph 4, line 65, does not end with a full stop.
Thank you.
Corrected.
5. Paragraph 5 needs to be improved. It contains a run-on sentence, and the study's objective is not clearly defined for the readers.
Correction were made according at the suggestion of Reviewer 2.
6. In the materials/methods section, the authors have not mentioned the analysis methods and conditions for FTIR, TGA, and DSC, which would aid reader understanding.
Thank you your suggestion. Necessary information h been added
7. Line 123: There seems to be extra spacing after the full stop.
Thank you.
Corrected.
8. Line 160: The abbreviation "app." for approximately is not defined earlier. All abbreviations should be defined upon their first appearance.
Abbreviation removed
9. Lines 175-176: The sentence could be rewritten, as starting a formal sentence with "But" in scientific articles is incorrect. Try using words like "however" or "nevertheless."
Thank you.
This sentence has been completely rewritten
10. Lines 180-181 need to be rewritten to sound more scientific.
These sentences have been rewritten
11. Figure 8b must be reinserted as it is an incomplete image.
This has been done
12. Lines 195-199 are right-aligned; the text should be adjusted for improved distribution.
Corrected.
13. In Figure 10(b), the legend color (light blue) does not match the graph legend notation (brown) for the 80:20 blend.
This error occurred as a result of an editorial reproduction of the original, where the color was correct (brown) we will try to fix this error.
14. Lines 263-265 need language improvement. "The picture reverses" does not fit in the scientific explanation.
This phrase has been completely rewritten.
